# China's Maritime Economic Development: A Review, the Future Trend, and Sustainability Implications

**Wai-Ming To [1]**  **and Peter K. C. Lee [2],\***

[1]   School of Business, Macao Polytechnic Institute, Macao, China; wmto@ipm.edu.mo
[2]   Department of Logistics and Maritime Studies, The Hong Kong Polytechnic University, Hong Kong, China
\*   Correspondence: peter.kc.lee@polyu.edu.hk; Tel.: +852-2766-7415

**Abstract:** China has experienced unprecedented economic growth and structural change in the past decades. This paper reviews the development of China's maritime economy for the period of 2002 to 2017. According to official data from China's government, the total production value of China's maritime economy increased from RMB 1068 billion in 2002 to RMB 7761 billion in 2017, thus contributing to about 10 percent of China's total gross domestic product. This paper applies four-parameter logistic models to identify the associated trends and predicts the near-future values for the total, primary, secondary, and tertiary sectors of China's maritime economy. It is projected that China's total maritime economy will reach RMB 8894 billion in 2019. Besides, the growth of China's maritime economy has been and will be dominated by the growth in the tertiary sector, particularly of China's coastal tourism and transport industries. Specifically, China's coastal tourism increased from RMB 147.8 billion in 2002 to RMB 1463.6 billion in 2017. In terms of spatial development, the east and north-east coasts of China experience more rapid maritime economic growth than the south-east coast. The implications for the sustainability of China's maritime economy are presented.

**Keywords:** maritime economy; China; sectoral analysis; geospatial analysis; sustainability implications

## 1. Introduction

Over 70 percent of the Earth's surface is water-covered and the majority of it is made up of seas and oceans. This is why humans have relied on seas and oceans for getting their food, obtaining other marine resources, exchanging goods and services between different geographical locations, and enjoying leisure activities. Thus, careful management of oceans and marine resources has been included as one of the United Nations' sustainable development goals [1]. The role of maritime-related activities in a country's economy can be substantial, particularly for island countries and countries in coastal areas. The European Commission indicated that about 41 percent of the European Union's (EU) population were living in coastal regions in 2011 and many EU economic activities could be considered as part of the EU's maritime economy [2]. The main maritime economic activities in the EU include tourism, transport, maritime manufacturing, fisheries, aquaculture, and fish processing. Studying the maritime economy in the United Kingdom (UK) from a historical perspective, Starkey and Jamieson [3] noted that its maritime economy has not been considered as a sector in the UK official statistical records, similar to other countries such as the United States [4]. Nevertheless, Starkey and Jamieson [3] suggested that the UK's maritime economy should cover a wide range of maritime-related industries and categorized those industries into four functional areas, i.e., the shipping, shipbuilding, and port industries, the marine fishery, offshore oil and gas industries, the maritime defense industry, and the growing coastal leisure industry. As one of the most advanced island countries in the world, the UK's maritime economy has played a significant role in its economic development in the past hundred forty

years [3]. Despite the paramount social and economic value of marine waters, there is scant research or literature on characterizing and measuring the maritime economy of specific countries or regions [5,6].

The Standardization Administration of China defined maritime economy as covering the exploitation, usage, and protection of oceanic resources and their associated activities [7]. China's maritime economy can be categorized into three sectors: primary, secondary, and tertiary. The primary sector includes marine fishery and aquaculture industries. The secondary sector includes the salt industry, the offshore oil and gas industry, the mining industry, the shipbuilding industry, and the pharmaceutical industry based on marine resources. The tertiary sector includes the transport industry and coastal tourism, and their associated service industry. Reviewing the development of maritime economy in China, Zhang [8] and Ma et al. [9] noted that China's maritime economy was first mentioned by China's economists Dixin Xu and Guangyuan Yu as a component of new economic indicators in the Meeting of National Planning Office of Philosophy and Social Science in 1978. Xu [10] reported that China's maritime economy was about RMB 8 billion in 1980 and increased to RMB 43.8 billion in 1990, and to RMB 413.3 billion in 2000. In the past twenty years, China's maritime economy has been studied in the fields of geographical science [9,11] and regional economics [12,13]. The State Oceanic Administration of China has published the annual report of China's maritime economic development since 2002 [14]. The report presents an overview of China's maritime economic development, its key compositions, and the regional contributions to China's maritime economy. Sun et al. [15] explored the growth of China's maritime economy during the period of 1996–2013. They reported that China's total maritime economy had grown continuously and Shanghai and Tianjin had experienced much faster development of the maritime economy than other provinces. More recently, Wang and Wang [16] studied the contribution of China's maritime sectors to China's total maritime economy using the input–output approach for the years 2002, 2007, and 2012, respectively. The input–output approach is able to reveal the inter-industry linkage effects by using the Leontief inverse matrix of an input–output table [16]. This approach has been used to study the forward linkage effect of different maritime sectors on the national economy in Korea [17,18] and Japan [19] for a particular year. However, there is still a lack of research studies on modeling China's maritime economic growth based on long-term time series and exploring the growth patterns and trends of the primary, secondary, and tertiary sectors of China's maritime economy. Wilkinson et al. [20] suggest that "long-term" means more than ten years, "medium-term" means three to ten years, and "short-term" means less than three years. Thus, the main purpose of the study is to address such gaps by applying logistic models to characterize China's total and sectoral maritime economic growth during the period of 2002 to 2017, and to project short-term growth in China's total maritime economy. Cubic polynomial models will be adopted to characterize the growth of the maritime economy in three different coastal regions.

## 2. Materials and Methods

This study uses the official data from the State Oceanic Administration of China (SOA, 2018) to profile the development of China's maritime economy for the period of 2002–2017. Besides, a four-parameter logistic model is applied to characterize the growth of China's maritime economy and its sub-sectors.

### 2.1. Logistic Model and Its Accuracy

A country's population, its gross domestic product (GDP), its sectoral added value, the use of a specific resource, and the adoption of a technology normally follow similar patterns, with a slow rate of increase in the beginning, followed by a period of accelerated growth, and then the rate of growth decreasing continuously until saturation when a specific parameter reaches a certain level. This type of S-shaped curve is known as the sigmoid or logistic function [21,22]. Rogers [21] noted that logistic curves are very common and exist in many social systems. Mahajan et al. [23] reviewed the modeling of new product acceptance in marketing literature and found that the logistic model and its variants

are applicable to a wide range of product categories across different industries. In the past ten years, logistic models have been used to characterize the adoption of management system standards across the world [24,25], the growth of Internet users in the US and China [26], the increase in electricity use in cities [27,28], and the growth of China's international sea freight [29]. Specifically, To and Lee [25,29] found that a four-parameter logistic model is a versatile approach to modeling macro-socioeconomic changes such as the adoption of generic management practices in firms at the global and regional levels and the increase in international sea freight due to changes in import and export activities in China since 1980. Following their approach, we model the growth of China's maritime economy and its sectors as:

$$Y(t) = Y_{initial} + \frac{Y_{increase}}{1 + e^{-\tau(t - t_{mid})}} \tag{1}$$

where $Y(t)$ is China's maritime economy (or its sectoral value added) in RMB at year $t$, $Y_{initial}$ is the baseline value, $Y_{increase}$ is the expected net increase in $Y$, $\tau$ is a growth coefficient, and $t_{mid}$ is the time (year) in which the maximum rate of growth would take place.

The accuracy of the logistic model is evaluated by three different measures. The first measure is the coefficient of determination, $R^2$. It represents how well the actual values are predicted by the regenerated data. The value of $R^2$ is between zero and one. A value of $R^2$ close to one means that the model has a high degree of accuracy. The formula of $R^2$ is given as:

$$R^2 = \frac{SSR}{SST} = 1 - \frac{SSE}{SST} \tag{2}$$

where $SSE$ is the sum of the squared error, $SSR$ is the sum of the squared regression, and $SST$ is the sum of the squared total.

The other two measures are the mean absolute error ($MAE$) and the mean absolute percentage error ($MAPE$) that determine the mean difference between the actual and predicted values in absolute terms and in percentage terms. The formula for $MAE$ is given as:

$$MAE = \frac{1}{n} \sum_{i=1}^{n} |e_i| \tag{3}$$

where $e$ is the difference between the actual and predicted values, and $n$ is the number of data points. The formula of $MAPE$ is given as:

$$MAPE = \frac{1}{n} \sum_{i=1}^{n} |p_i| \tag{4}$$

where $p$ is the percentage difference between the actual and predicted values, and $n$ is the number of data points.

### 2.2. Data Collection

China's total maritime economy, its sectoral components, and the contributions from three coastal regions including Bohai Bay Area, the Yangtze River Delta, and the Pearl River Delta were obtained from the annual reports published by the State Oceanic Administration of China for the period of 2002–2017 [14]. It should be noted that the State Oceanic Administration of China has published annual reports since 2002. According to the classification of China's coastal administrative areas, the Bohai Bay Area mainly covers several coastal areas such as the municipality of Tianjin, and the provinces of Liaoning, Hebei, and Shandong [14]. The Yangtze River Delta includes the municipality of Shanghai, and the provinces of Jiangsu and Zhejiang while the Pearl River Delta refers to cities in Guangdong such as Guangzhou, Shenzhen, Zhuhai, etc. Figure 1 shows the locations of these three coastal regions.

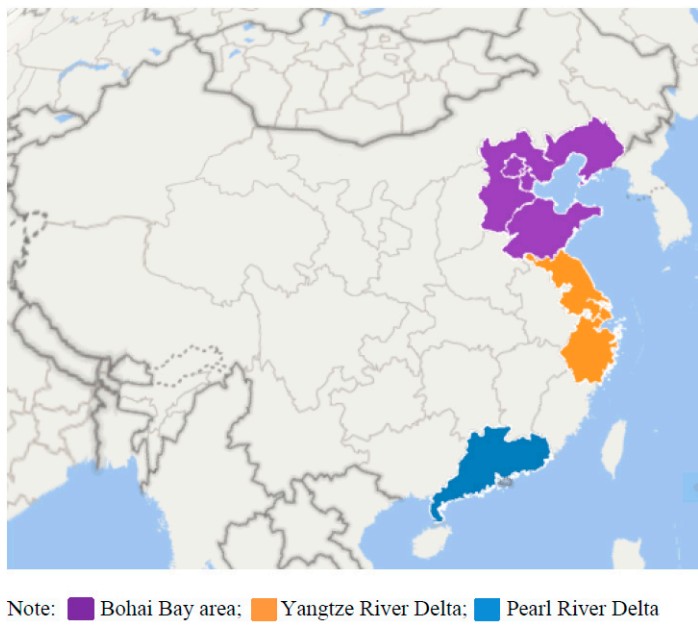

Note: ■ Bohai Bay area; ■ Yangtze River Delta; ■ Pearl River Delta

**Figure 1.** China's three main coastal regions.

## 3. Results

Table 1 presents details of China's maritime economy and its sectoral components for the period of 2002 to 2017. It was found that China's maritime economy increased from RMB 1067.7 billion in 2002, to RMB 3842.9 billion in 2010, then to RMB 7761.1 billion in 2017. During this time period, the primary sector increased by over four times from RMB 82.5 billion in 2002 to RMB 360 billion in 2017. The secondary sector increased by over six times from RMB 461.1 billion in 2002 to RMB 3009.2 billion in 2017 while the tertiary sector increased by over eight times from RMB 524.1 billion in 2002 to RMB 4391.9 billion in 2017.

**Table 1.** The size of China's maritime economy for the period of 2002 to 2017.

| Year | Production Value in Billion RMB | | | |
| --- | --- | --- | --- | --- |
| | Primary | Secondary | Tertiary | Total |
| 2002 | 82.5 | 461.1 | 524.1 | 1067.7 |
| 2003 | 90.7 | 539.2 | 574.5 | 1204.4 |
| 2004 | 93.7 | 645.8 | 690.2 | 1429.7 |
| 2005 | 103.3 | 852.2 | 847.1 | 1802.6 |
| 2006 | 110.5 | 985.8 | 999.5 | 2095.8 |
| 2007 | 127.4 | 1150.3 | 1215.2 | 2492.9 |
| 2008 | 160.8 | 1402.6 | 1402.8 | 2966.2 |
| 2009 | 187.9 | 1506.2 | 1502.3 | 3196.4 |
| 2010 | 206.7 | 1811.4 | 1825.8 | 3843.9 |
| 2011 | 232.7 | 2183.5 | 2140.8 | 4557.0 |
| 2012 | 268.3 | 2298.2 | 2442.2 | 5008.7 |
| 2013 | 291.8 | 2490.8 | 2648.7 | 5431.3 |
| 2014 | 322.6 | 2704.9 | 2966.1 | 5993.6 |
| 2015 | 329.2 | 2749.2 | 3388.5 | 6466.9 |
| 2016 | 356.6 | 2848.8 | 3845.3 | 7050.7 |
| 2017 | 360.0 | 3009.2 | 4391.9 | 7761.1 |

When the contribution of each sector to China's total maritime economy was characterized in terms of percentage, it was found that the primary sector decreased almost continuously from 7.7 percent in 2002 to 4.6 percent in 2017 while the tertiary sector changed from 49.1 percent in 2002 to 56.6 percent in 2017, as shown in Figure 2.

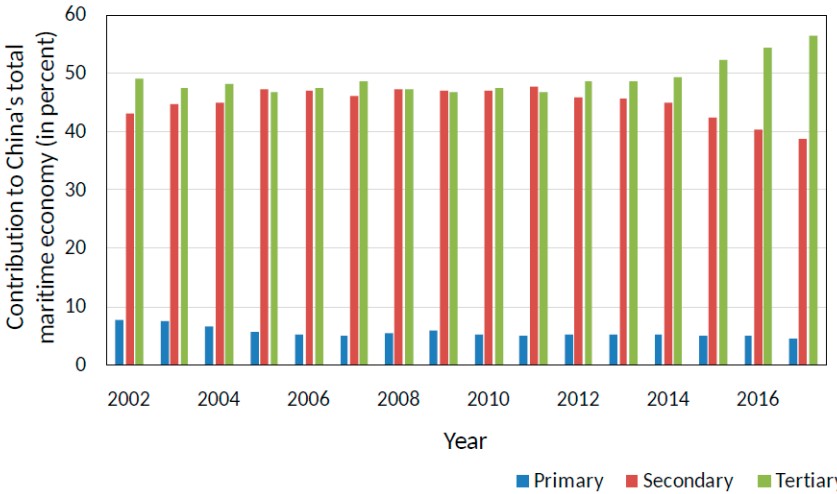

**Figure 2.** Contribution from China's sectoral maritime economy from 2002 to 2017.

### 3.1. Modeling and Trends of China's Maritime Economy

China's total maritime economy has grown rapidly over the past fifteen years and the rate of growth is likely to slow down in the coming years. A four-parameter logistic model was applied to the time series of China's total maritime economic data from 2002 to 2017. The following was obtained using the nonlinear algorithm as suggested by Motulsky and Christopulos [30]:

$$Y_{total}(t) = 260 + \frac{13,700}{1 + e^{-0.19(t-2016)}} \tag{5}$$

where $Y_{total}(t)$ is China's total maritime economy in billion RMB at year $t$, the baseline value is 260 billion RMB, the expected net increase in China's total maritime economy is 13,700 billion RMB, the growth coefficient is 0.19, and the time (year) in which the maximum rate of growth should have taken place is 2016. Figure 3 shows China's total maritime economy and the predicted value based on Equation (5) for the years from 2002 to 2017. The $R^2$ value for the actual and predicted total maritime economy was 0.993. The values of *MAE* and *MAPE* were 152 billion RMB and 4.9 percent, respectively.

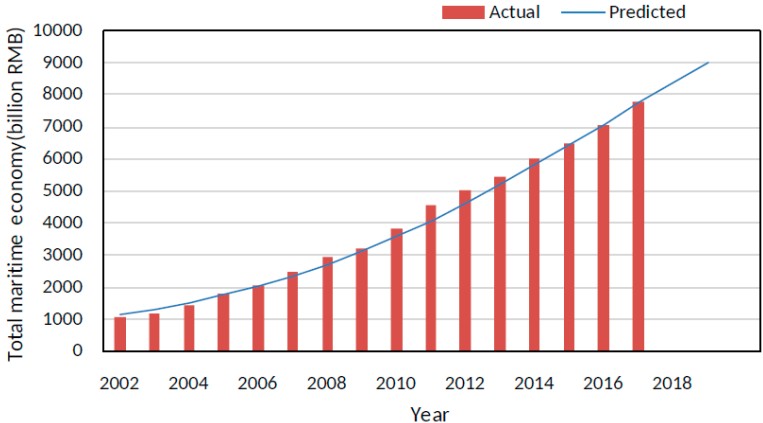

**Figure 3.** China's total maritime economy from 2002 to 2017.

China's maritime economy was disaggregated into three sub-sectors, namely the primary, the secondary, and the tertiary sectors. The four-parameter logistic model was applied to each of the time series of three China's sectoral maritime economic data from 2002 to 2017. The resulting

model equations are given below. Table 2 presents $R^2$ values of the identified equations, and the values of *MAE* and *MAPE*.

$$Y_{primary}(t) = 70 + \frac{330}{1 + e^{-0.36(t-2011)}} \tag{6}$$

$$Y_{secondary}(t) = 150 + \frac{3300}{1 + e^{-0.28(t-2010)}} \tag{7}$$

$$Y_{tertiary}(t) = 100 + \frac{9300}{1 + e^{-0.19(t-2018)}} \tag{8}$$

**Table 2.** Sectoral logistic models, $R^2$, *MAE*, and *MAPE* values.

| Sector | Logistic Model | $R^2$ | *MAE* (Billion RMB) | *MAPE* (Percent) |
|---|---|---|---|---|
| Primary | Equation (6) | 0.997 | 4 | 2.2 |
| Secondary | Equation (7) | 0.996 | 45 | 2.8 |
| Tertiary | Equation (8) | 0.996 | 44 | 3.2 |

Figure 4 shows the values of China's primary maritime sector and the predicted values based on the first equation given in Table 2. It was found that the four-parameter logistic model was able to reproduce the sectoral maritime economy with acceptable accuracy.

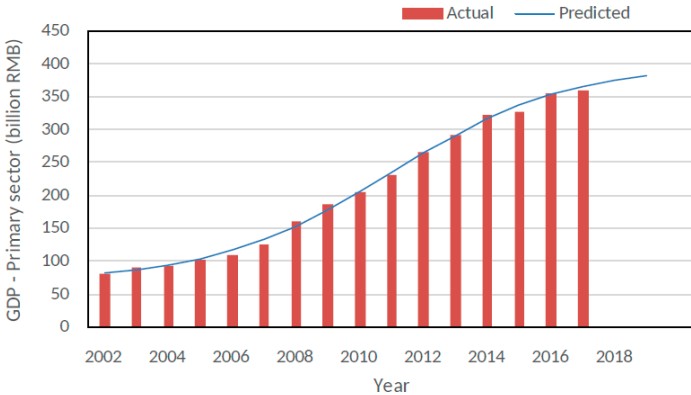

**Figure 4.** China's primary maritime sector from 2002 to 2017.

Figures 5 and 6 show the values of China's secondary and tertiary maritime sectors and the predicted values based on the second and third equations given in Table 2. It was found that four-parameter logistic models could reproduce the values of both China's secondary and tertiary maritime sectors quite accurately.

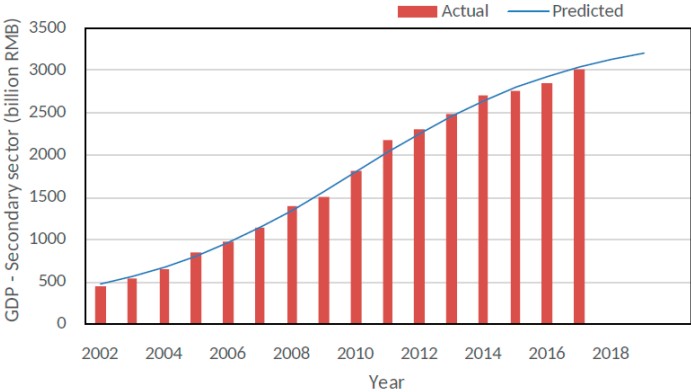

**Figure 5.** China's secondary maritime sector from 2002 to 2017.

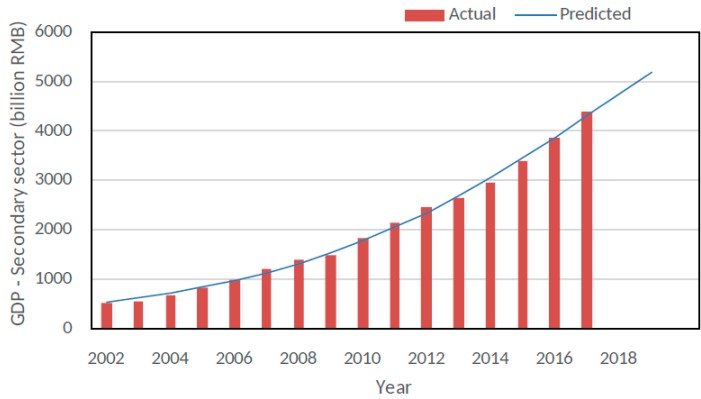

**Figure 6.** China's tertiary maritime sector in the period 2002 to 2017.

It was possible to predict the short-term (i.e., two years) growth of China's maritime economy based on Equation (5) and the ones shown in Table 2. Table 3 shows that China's predicted total maritime economy will be 8396 billion RMB in 2018 and 9011 billion RMB in 2019, respectively, based on Equation (5). When the sectoral economic models i.e., Equations (6)–(8) were used, China's predicted total maritime economy could be 8258 billion RMB in 2018 and 8777 billion RMB in 2019, respectively. Hence, it can be estimated that China's total maritime economy was approximately 8327 (=0.5 × (8396 + 8258)) billion RMB in 2018 and approximately 8894 (=0.5 × (9011 + 8777)) billion RMB in 2019.

**Table 3.** China's predicted maritime economy in billion RBM in 2018 and 2019.

| Year | Actual | Overall | Sectoral | | | |
|------|--------|---------|----------|----------|----------|-------|
| | | Equation (5) | Equation (6) | Equation (7) | Equation (8) | Total |
| 2017 | 7761 | 7759 | 366 | 3042 | 4310 | 7718 |
| 2018 | | 8396 | 375 | 3133 | 4750 | 8258 |
| 2019 | | 9011 | 382 | 3205 | 5190 | 8777 |

*3.2. Changes in China's Maritime Industries in the past Fifteen Years*

In 2002, China's total maritime economy was 1067.7 billion RMB and there were three categories of industries exceeding 100 billion RMB. They were coastal tourism with 147.8 billion RMB, followed by marine fishery, aquaculture, and their associated services with 120.8 billion RMB, and maritime transport with 103.9 billion RMB. In 2017, China's total maritime economy increased to 7761.1 billion RMB and there were seven categories of industries exceeding 100 billion RMB. Specifically, China's coastal tourism increased by some 10 times to 1463.6 billion RMB in 2017. The second largest maritime industry in 2017 was maritime transport, which contributed to 631.2 billion RMB to China's GDP in 2017, representing an increase of 103.9 billion RMB since 2002. The third largest category in 2017 was marine fisheries, aquaculture, and their associated services industries. This category of industry expanded steadily from 120.8 billion RMB in 2002 to 467.6 billion RMB in 2017. The other four industries including the marine engineering architecture, marine shipbuilding, offshore oil and gas, and marine chemical industries contributed between 104 billion RMB and 184 billion RMB to China's GDP in 2017. Figure 7 shows changes in key Chinese maritime industries in the period 2002 to 2017.

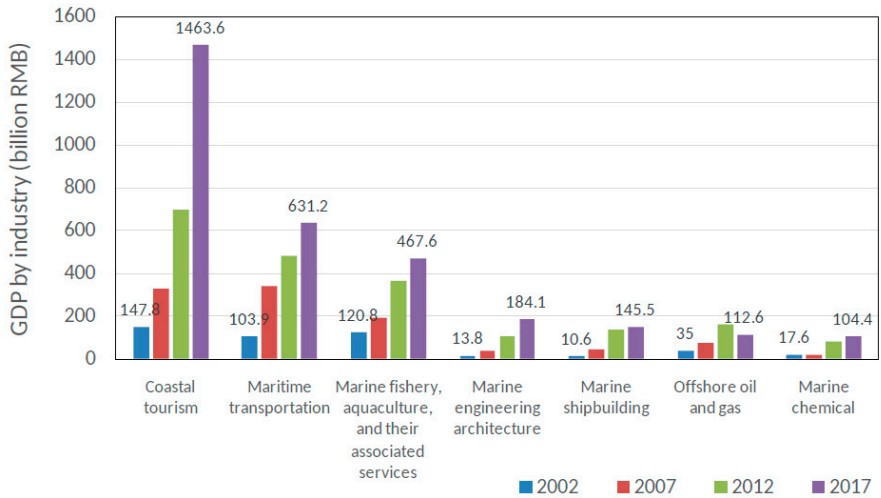

**Figure 7.** Changes in China's key maritime industries between 2002 and 2017.

### 3.3. Contributions to China's Maritime Economy from Three Coastal Regions

China's total maritime economy expanded from RMB 8 billion in 1980 to RMB 413.3 billion in 2000 [10]. In terms of provincial contributions, Guangdong has been ranked first since China opened up its economy in 1978, followed by Shangdong. Region-wise, Bohai Bay Area had the largest contribution to China's maritime economy in the late 1990s. It was surpassed by Yangtze River Delta between 2002 and 2005. Bohai Bay Area has the largest contribution to China's maritime economy again since 2006. Figure 8 shows the contributions to China's maritime economy from the three coastal regions.

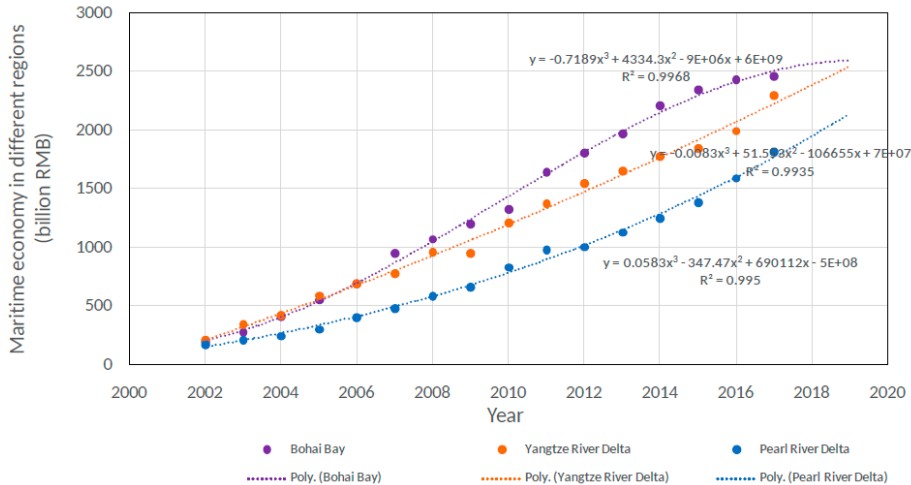

**Figure 8.** Contributions to China's maritime economy from three coastal regions from 2002 to 2017.

Figure 8 shows that the gap between the contributions from the Bohai Bay Area and the Yangtze River Delta is likely to decrease in the coming two years. It is projected that the maritime industries in the Bohai Bay Area and the Yangtze River Delta would produce over 2500 billion RMB each and in the Pearl River Delta would produce about 2150 billion RMB in 2019.

### 3.4. A Model Linking China's Maritime Economy to Its Sustainability Implication

The continual development of China's maritime economy inevitably requires the exploitation of more ecological resources [31] and produces more pollutants, such as greenhouse gases [29]. Specifically, China's total fish production is one of the major maritime industries in China because it employs more than 14 million people in China [31]. Yet, China's total fish production had increased by more than 82.6 percent from 38.3 million tonnes in 2002 to 70.0 million tonnes in 2017 [31]. Figure 9

shows that China's maritime economy plotted against China's total fish production during the period of 2002–2017. Mathematically speaking, the linear model between China's maritime economy and total fish production is given as follows:

$$y_{total\ fish\ production} = 0.0047 x_{maritime\ economy} + 34.127, \tag{9}$$

where $y_{total\ fish\ production}$ is China's total fish production in million tonnes and $x_{maritime\ economy}$ is China's total maritime economy in billion RMB. The $R^2$ value between China's total fish production and China's total maritime economy was 0.9977, meaning that more than 99.77 percentage change in the variance of China's total fish production could be explained by the change in China's total maritime economy. Equation (9) implies that a change of 1000 billion RMB will lead to an increase of 4.7 million tonnes of total fish production. As China's total maritime economy would increase by 1133 billion RMB in from 2017 to 2019 (see Section 3.1), it is expected that this change will lead to China's total fish production increasing by another 5.3 million tonnes. This potential change in total fish production, on one hand, will cause a greater ecological impact, and, on the other hand, can further strain China's relationship with its neighboring countries such as Japan, Korea, Vietnam, and Thailand due to fishing in open seas [32].

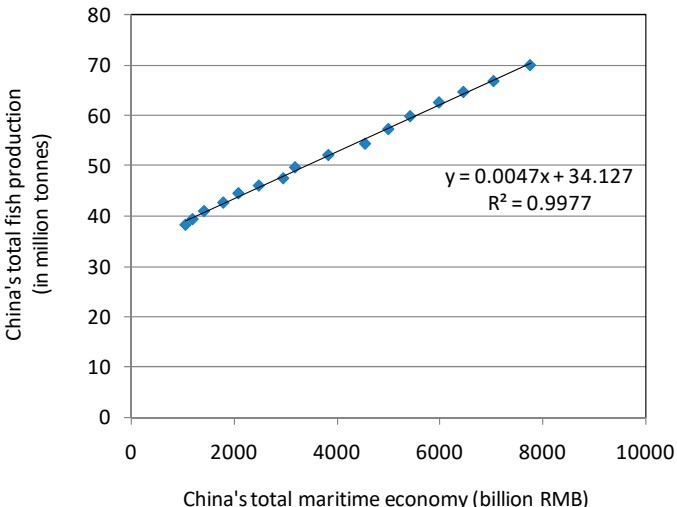

**Figure 9.** Change in China's total fish production vs. total maritime economy from 2002 to 2017.

## 4. Discussion

China's total maritime economy has been modeled in this paper using a four-parameter logistic model in which the maximum rate of growth had taken place in 2016. When the sectoral maritime economies are separately analyzed, it is found that the contribution from the primary sector to China's total maritime economy has decreased from 7.7 percent to 4.6 percent over the past fifteen years. The maximum rate of growth of China's primary maritime sector had taken place in 2011. The contribution from the secondary sector to China's total maritime economy was 43.2 percent in 2002, increasing to over 47 percent in 2005 and then decreasing continuously from 47.9 percent in 2011 to 38.8 percent in 2017. The application of the four-parameter logistic model has shown that the maximum rate of growth of China's secondary maritime sector occurred in 2010. The contribution from the tertiary sector to China's total maritime economy ranged between 46 percent and 49 percent during the period 2002 and 2013. Since 2013, its contribution has been increasing continuously to reach 56.6 percent in 2017. The four-parameter logistic model, however, showed that the maximum rate of growth of China's tertiary maritime sector would most likely take place in 2018. The four-parameter logistic models of China's total and sectoral maritime economy predict that China's total maritime economy would reach 8327 and 8894 billion RMB in 2018 and 2019, with the annual growth rate

being at 7.3 percent and 6.8 percent in 2018 and 2019, respectively. These predictions are based on the assumption that there is no structural break in the underlying economic condition, i.e., no catastrophic economic crisis in China. These figures are also much higher than that of the EU maritime economy based the gross value added at EUR 174.2 billion, i.e., 2.3% of the total EU GDP in 2016 [33] and the US maritime economy at about USD 320 billion, i.e., 1.8 percent of the total US GDP in 2015 [34].

*Sustainability Implications*

China's maritime economy could be characterized by several key maritime industries as shown in Figure 7. Specifically, it was found that coastal tourism grew substantially from 147.8 billion RMB in 2002 to 1463.6 billion RMB in 2017. This suggests that China's government should pay great attention to both the positive and negative effects on the economic, social, and environmental aspects of coastal regions and municipalities. This is because, coastal tourism, on the one hand, increases a city/region's gross domestic product, creates jobs, improves the city/region's image, fosters cultural exchange, and encourages infrastructure development [35,36]. On the other hand, coastal tourism adversely affects the biophysical environment. The impacts include the increase in air and noise pollution from increased traffic, the increase in water pollution and waste, and the adverse change in wetlands and rivers due to a wide range of tourism-related commercial activities, and the decrease of fish population due to recreational fishing [35–39]. The second maritime industry contributing significantly to China's total maritime economy is maritime transport. As China has been the world's top exporting country since 2005 and the second top importing country since 2009 [40] and the majority of import–export trade relies on maritime transport [29,41], the steady growth of maritime transport including the development of port and logistics facilities helps stabilize China's economic and social development. On the other hand, maritime transport produces a significant amount of greenhouse gases and other pollutants [29,31,41]. Specifically, China's international sea freight—a subsector of China's maritime economy—and its greenhouse gases emissions have followed logistic growth in the past decades [29]. Thus, it is crucial that better demand-side management, more efficient up-stream logistics services, minimization of road and port congestion, reductions in the idling time of container and cargo ships at ports, and use of cleaner fuels and better engines can mitigate the environmental problems associated with maritime transport. The third important category of maritime industries includes maritime fishery and aquaculture and their associated service industries. According to the Food and Agriculture Organization of the United Nations [31], China has been one of the world's top fish product producers since 2002, followed by India and Indonesia. The amount of fish production was 38.3 million tonnes in 2002, with 14.2 million tonnes (37%) from capture and 24.1 million tonnes (63%) from aquaculture. Likewise, its fish production increased to 70.0 million tonnes in 2017, with 16.2 million tonnes (23%) from capture and 53.7 million tonnes (77%) from aquaculture [31]. Thus, the compound annual growth rate of marine capture was 0.9 percent and that of aquaculture was 5.5% during the period 2002 to 2017. In terms of social function, China's maritime fishery, aquaculture, and other services provided jobs for over 14 million people in 2017 [31]. Yet, expansion of the maritime fishery and aquaculture may not be sustainable if the exploitation of these resources exceeds their regeneration capabilities, i.e., the maximum sustainable yields [42]. Because of this, China, as one of the first six nations ratifying the Convention on Biological Diversity, established its National Biodiversity Conservation Action Plan in 1994 [43]. It also revised laws such as the Marine Environment Protection Law and Fisheries Law and implemented paid use system to better protect and improve the marine environment and conserve marine resources [43–47]. Specifically, Mu et al. [45] showed that China's marine catch had increased sharply from about 3.8 Mt in 1985 to 11 Mt in 1995. However, China's National Biodiversity Conservation Action Plan slowed down the growth of marine catch significantly, causing the total marine catch increasing slowly to 14.2 Mt in 2002 [41,45]. Besides, the overexploitation and the degradation of fishery habitats have caused grave concern for both China's government and fishermen [45]. Thus, China imposes an annual fishing moratorium between 1 May and 16 August in South China Sea, East China Sea, Yellow Sea, and Bohai Sea [45]. Still, a more holistic approach

needs to be employed to balance the ecological, economic, and social aspects of China's marine fishery resources.

## 5. Conclusions

Ocean and coastal management have played a crucial role in the economic and sustainable development of developing countries, particularly China, over the past two decades. However, to our knowledge, there have been few prior studies exploring China's total and sectoral maritime economic growth based on long-term (more than ten years) time series and predicting their short-term (two years) trends. This paper has identified that China's total maritime economy and its sectoral economies could be characterized by four-parameter logistic models. China's total maritime economy, its primary component and secondary component have already passed their maximum growth rate. Actually, its tertiary component reached its maximum growth rate in 2018. Nevertheless, China's total maritime economy will maintain an annual growth of around 7 percent in 2018 and 2019. In the near future, China's maritime economy will be dominated by growth in the tertiary sector, particularly the coastal tourism and transport industries. Our study has also found that the Bohai region and the Yangtze River Delta region witnessed a more rapid maritime economic growth than the south-east coast of China i.e., the Pearl River Delta region during the period of 2002–2015.

**Author Contributions:** Conceptualization, W.-M.T. and P.K.C.L.; methodology, W.-M.T. and P.K.C.L.; formal analysis, W.-M.T.; writing—original draft preparation, W.-M.T.; writing—review and editing, P.K.C.L.; project administration, W.-M.T.; funding acquisition, P.K.C.L.

**Funding:** This research received no external funding.

**Acknowledgments:** This research was supported in part by the Hong Kong Polytechnic University (Project Code: G-UADS).

**Conflicts of Interest:** The authors declare no conflict of interest.

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
