# Peer review of "China’s Maritime Economic Development: A Review, the Future Trend, and Sustainability Implications"

_sustainability, doi:10.3390/su10124844_

Round 1

Reviewer 1 Report

The article China´s Maritime Economic Development : A Review, the Future Trends, and Sustainability Implications is interesting topic connected with economic growth of China economy. I think the authors can  show more results in the abstract. In introduction the authors describe the importance of management of oceans and marine resources. I recommend  better connection in the introduction with the goal and the topic of the article.

May be the different approaches or comparison of access to this could eb define.

 In row 73 the authors mentioned the years 2002-2015, the rest of the article is about  the years 2002 - 2017. 

In the prediction  authors use only the trends fromthe model, did they add some economical expectancy?

In Discussion the authors could compare the results with the global point of view on the  maritime economy and copare with EU or USA.

Author Response

Reviewer 1

1st Comment: The article China´s Maritime Economic Development: A Review, the Future Trends, and Sustainability Implications is interesting topic connected with economic growth of China economy. I think the authors can show more results in the abstract.

Our response: Thanks very much for your comment. As suggested by you, we showed more results in the section of Abstract as:

“…It is projected that China’s total maritime economy will reach RMB 8,894 billion in 2019. Besides, the growth of China’s maritime economy has been and will be dominated by the growth in the tertiary sector, particularly of China’s coastal tourism and transport industries. Specifically, China’s coastal tourism increased from RMB 147.8 billion in 2002 to RMB 1,463.6 billion in 2017. In terms of spatial development, the east and north-east coasts of China experience…”

2nd Comment:  In introduction the authors describe the importance of management of oceans and marine resources. I recommend better connection in the introduction with the goal and the topic of the article.

Our response: In order to better connect the section of Introduction with the goal and the topic of the article (as suggested by you), we added the following sentence at the end of the 1st paragraph of Introduction:

“…Despite the paramount social and economic value of marine waters, there is scant research and literature on characterizing and measuring a country’s or regional maritime economy [5,6].”

The following sentences were added to the 2nd paragraph of Introduction:

“…Sun et al. [15] explored the growth of China’s maritime economy during the period of 1996-2013. They reported that China’s total maritime economy had growth continuously and Shanghai and Tianjin had experienced much faster development of the maritime economy than other provinces. More recently, Wang and Wang [16] studied the contribution of China’s maritime sectors to China’s total maritime economy using the input-output approach for the years 2002, 2007, and 2012, respectively. The input-output approach is able to reveal the inter-industry linkage effects by using the Leontief inverse matrix of an input-output table [16]. This approach has been used to study the forward linkage effect of different maritime sectors on the national economy in Korea [17,18] and Japan [19] for a particular year. However, there is still a lack of research studies on modeling China’s maritime economic growth based on long-term time series and exploring the growth patterns and trends of the primary, secondary, and tertiary sectors of China’s maritime economy…”

Because of these changes, the following seven references were added:

5.       Fernández-Macho, J.; Murillas, A.; Ansuategi, A.; Escapa, M.; Gallastegui, C.; González, P.; Prellezo, R.; Virto, J. Measuring the maritime economy: Spain in the European Atlantic Arc. Mar. Policy 2015, 60, 49-61.

6.        Fernández-Macho, J.; González, P.; Virto, J. An index to assess maritime importance in the European Atlantic economy. Mar. Policy 2016, 64, 72-81.

15.      Sun, C.; Li, X.; Zou, W.; Wang, S; Wang, Z. Chinese marine economy development: Dynamic evolution and spatial difference. Chinese Geogr. Sci. 2018, 28(1), 111-126.

16.      Wang, Y.; Wang, N. The role of the marine industry in China’s national economy: An input–output analysis. Mar. Policy 2019, 99, 42-49.

17.      Kwak, S.J.; Yoo, S.H.; Chang, J.I. The role of the maritime industry in the Korean national economy: An input–output analysis. Mar. Policy 2005, 29(4), 371-383.

18.      Lee, M.K.; Yoo, S.H. The role of the capture fisheries and aquaculture sectors in the Korean national economy: An input–output analysis. Mar. Policy 2014, 44, 448-456.

19.      Nakahara, H. Economic contribution of the marine sector to the Japanese Economy. Trop. Coast 2009, 16(1), 49-53.

3rd Comment: May be the different approaches or comparison of access to this could be defined.

Our response: Thanks very much for your comment and suggestion. We added the following sentences that describe the input-output method for identifying the inter-industry linkage effects that may have impact on the upstream and downstream economic activities in Introduction.

“…More recently, Wang and Wang [16] studied the contribution of China’s maritime sectors to China’s total maritime economy using the input-output approach for the years 2002, 2007, and 2012, respectively. The input-output approach is able to reveal the inter-industry linkage effects by using the Leontief inverse matrix of an input-output table [16]. This approach has been used to study the forward linkage effect of different maritime sectors on the national economy in Korea [17,18] and Japan [19] for a particular year…”

4th Comment:  In row 73 the authors mentioned the years 2002-2015, the rest of the article is about the years 2002 - 2017.

Our response: Thanks very much for your comment. It was a typing mistake and we corrected it to 2002-2017 in the revised manuscript.

5th Comment: In the prediction authors use only the trends from the model, did they add some economical expectancy?

Our response: Thanks very much for your comment. The logistic model works well in most economic development including the development of the maritime economy. However, if a catastrophic economic crisis happens, it may lead to a significant drop i.e. structural break in the predicted values. Hence, we rewrote the sentences in Discussion as:

“…The four-parameter logistic models of China’s total and sectoral maritime economy predict that China’s total maritime economy would reach 8,327 and 8,894 billion RMB in 2018 and 2019, with the annual growth rate being at 7.3 percent and 6.8 percent in 2018 and 2019, respectively. These predictions are based on the assumption that there is no structural break in the underlying economic condition, i.e. no catastrophic economic crisis in China.”

5th Comment: In Discussion the authors could compare the results with the global point of view on the maritime economy and compare with EU or USA.

Our response: Thanks very much for your comment and suggestion. We obtained the latest reports from the European Commission and the US Government. As suggested by you, we added the following sentence at the end of Discussion.

“…. These figures are also much higher than that of the EU maritime economy based the gross value added at EUR 174.2 billion, i.e. 2.3% of the total EU GDP in 2016 [31] and the US maritime economy at about USD 320 billion, i.e. 1.8 percent of the total US GDP in 2015 [32].”

The following two references were added:

31.          The European Commission. What is the Blue Economy? The European Commission's Directorate General for Maritime Affairs and Fisheries, Brussel, Belgium, 2017.

32.          The US National Ocean and Atmospheric Administration. NOAA Report on the U.S. Ocean and Great Lakes Economy. The US Office for Coastal Management, South Carolina, the USA, 2018.

Dear Reviewer 1 - please accept our sincere thanks for your valuable comments and suggestions that improve the way we presented the study substantially.

Reviewer 2 Report

Thank you for your work, here are some suggestions,

1.      I understand that you made an effort of demonstrating sustainability implications in your research, however, it was not reflected in your model construction and data, which means there is nothing empirical to support your conclusion and discussion, especially the one you made on sustainability. Therefore, economic development is your main priority instead of sustainability, is it?

2.      In your sustainability implications part from line 245, you discussed tertiary industry, secondary industry and primary industry separately, and implies that they shed both positive and negative effects over sustainable development, which is true but lack of originality and also it does not related to your empirical analysis, I am wondering, what conclusions can you make based on your model and data analysis, could it be related to sustainability?

3.      Line 119 do you mean Shandong Province?

Author Response

Reviewer 2

 1st Comment:  I understand that you made an effort of demonstrating sustainability implications in your research, however, it was not reflected in your model construction and data, which means there is nothing empirical to support your conclusion and discussion, especially the one you made on sustainability. Therefore, economic development is your main priority instead of sustainability, is it?

Our response: Thanks so much for your comment. You are right. We focused more on the economic side of China’s maritime economy due to the information available from the State Oceanic Administration of China (our Reference 14 in the revised manuscript). However, it is a fact that a country’s maritime-related activities will not only bring economic benefits, it has significant implications on the utilization of natural resources such as fisheries, land (for ports and coastal tourism), and energy sources (for transport and electricity consumption). As we stated in Introduction:

“              Over 70 percent of the Earth’s surface is water-covered and the majority of it is made up of seas and oceans. This is why humans have relied on seas and oceans for getting their food, obtaining other marine resources, exchanging goods and services between different geographical locations, and enjoying leisure activities. Thus, careful management of oceans and marine resources has been included as one of the United Nations’ sustainable development goals [1]. The role of maritime-related activities in a country’s economy can be substantial,…”

The model we presented illustrated the growth of China’s maritime economy in terms of RMB. It is closely related to the use of natural resources. Thus, we rewrote the implications in Section 4.1 as:

“…As China has been the world’s top exporting country since 2005 and the second top importing country since 2009 [38] and the majority of import-export trade relies on maritime transport [29,39], the steady growth of maritime transport including the development of port and logistics facilities helps stabilize China’s economic and social development. On the other hand, maritime transport produces a significant amount of greenhouse gases and other pollutants [29,39,40]. Specifically, China’s international sea freight – a subsector of China’s maritime economy and its greenhouse gases emissions have followed logistic growth in the past decades [29]. Thus, it is crucial that better demand side management, more efficient up-stream logistics services, minimization of road and port congestion, reductions in the idling time of container and cargo ships at ports, and use of cleaner fuels and better engines can mitigate the environmental problems associated with maritime transport. The third important category of maritime industries includes the maritime fishery, aquaculture, and their associated services industries. According to the Food and Agriculture Organization of the United Nations [41], China has been one of the world’s top fish product producers since 2002, followed by India and Indonesia. The amount of fish production was 38.3 million tonnes in 2002, with 14.2 million tonnes (37%) from capture and 24.1 million tonnes (63%) from aquaculture. Likewise, its fish production increased to 65.2 million tonnes in 2015, with 17.6 million tonnes (27%) from capture and 47.6 million tonnes (73%) from aquaculture [41]. Thus, the compound annual growth rate of marine capture was 1.4 percent and that of aquaculture was 4.6% during the period 2002 to 2015…”

In fact, we explored the growth of China’s international sea freight (a subsector of China’s maritime economy) and found that it followed the logistic model as what we reported in the manuscript for China’s total maritime economy. We published the article as “Reference 29 “:

29.          To, W. M.; Lee, P. K. C. GHG emissions from China's international sea freight transport: A review and the future trend. Int. J. Ship. Trans. Log. 2018, 10(4), 455-467.

2nd Comment: In your sustainability implications part from line 245, you discussed tertiary industry, secondary industry and primary industry separately, and implies that they shed both positive and negative effects over sustainable development, which is true but lack of originality and also it does not related to your empirical analysis, I am wondering, what conclusions can you make based on your model and data analysis, could it be related to sustainability?

Our response: Thanks so much for your comment. We agreed that we used the model to illustrate the trend and potential growth of China’s total and sectoral maritime economy (based on the information from the State Oceanic Administration of China). However, although the models of China’s and sectoral economic growth in the maritime economy cannot be translated directly to the use of natural resources (due to its complexity and lack of data from the official site of China’s government), they have significant implications on the use and conservation of resources in China. That is the reason we used the terms “sustainability implications” and we tried to obtain more information (historical ones) from various sources such as some related articles and the reports from the World Bank and the Food and Agriculture Organization of the United Nations (see our References 29, 38 and 41).  At least, they reported the growth of some China’s maritime sectors in a number of years. We sincerely hope that you understand our difficulties in obtaining more resource-related data.

3rd Comment: Line 119 do you mean Shandong Province?

Our response: Thanks so much for your comment. You are correct. We made a typing mistake and we corrected it to “Shandong” in the revised manuscript.

Dear Reviewer 2 - please accept our sincere thanks for your comments and suggestions that help us improve the manuscript continually.

Reviewer 3 Report

This paper concisely deals with a relevant topic often discussed in economics and economic development (i.e. the maritime economy in relation to overall economic growth via different mechanisms -trade, transport, fishing, shipping, etc). This paper, however, explores exclusively the composition of the sector in China since 2002 and relates the findings to potential sustainability concerns.  Given the seemingly importance of this sector to the whole Chinese economic growth, the article would be of interest to a larger audience. 

There are, however, a few aspects that should be addressed to make the paper more relevant:

-      The introduction’s focus on the EU and UK, while interesting, is not clearly linked to the Chinese. Indeed, it serves the purpose of arguing that the maritime economy is important and should be studied more, but it is not clear how, for instance, the claims by Starkey and Jamieson on the functional areas relate to the Chinese case. 

-      The authors need to explain why they have chosen 2002 as the starting point for the analysis, since they have data on 1980 (section 3.3.) after the Chinese economy open. Equally important, the changes in importance of the regions’ economic contributions are left unexplored. 

-      While the method used seems to fit well with the evolution of Chinese maritime economy, the authors would benefit by discussing this approach in relation to others used in the literature or to previous work on the Chinese case (for instance, input-output analysis, previous research on Chinese maritime sector and to other cases, for instance Korea or Japan as relevant neighbouring countries). 

-      The sustainability implications are drawn from secondary sources. It would be good to know whether the discussion is based on hypothetical implications or actual findings for China’s maritime sector. 

-      In relation to the conclusions, if the authors want to relate their findings to management, more should be known about the actual policies to promote the changes. Relatedly, that would bring more insights onto causes of growth and hence more robust evidence on the strength of their predictions. Would the authors indicate that the sustainability of the sector is exclusively dependant on natural/ecological constraints?  

The paper is well written and structured. It describes the changes in composition of the sector and attempts to predict. It might benefit from setting some expectations prior to the analysis as to make the discussion of the implications more insightful. 

Author Response

Reviewer 3

1st Comment: This paper concisely deals with a relevant topic often discussed in economics and economic development (i.e. the maritime economy in relation to overall economic growth via different mechanisms -trade, transport, fishing, shipping, etc). This paper, however, explores exclusively the composition of the sector in China since 2002 and relates the findings to potential sustainability concerns.  Given the seemingly importance of this sector to the whole Chinese economic growth, the article would be of interest to a larger audience.

Our response: Thanks very much for your comment and understanding the goal of our study.

2nd Comment: There are, however, a few aspects that should be addressed to make the paper more relevant:

-   The introduction’s focus on the EU and UK, while interesting, is not clearly linked to the Chinese. Indeed, it serves the purpose of arguing that the maritime economy is important and should be studied more, but it is not clear how, for instance, the claims by Starkey and Jamieson on the functional areas relate to the Chinese case.

Our response: Thanks very much for your comment and suggestion. In the revised manuscript, we rewrote the categorization offered by Starkey and Jamieson and put more emphasis on the growing coastal leisure (or tourism) industry which has also been experienced in China. We also added the following sentence to establish a close link between the 1st paragraph and the 2nd paragraph of Introduction that focuses on China’s maritime economy.

“…Starkey and Jamieson [3] suggested that UK’s maritime economy should cover a wide range of maritime-related industries and categorized those industries into four functional areas, i.e. the shipping, shipbuilding, and port industries, the marine fishery, offshore oil and gas industries, the maritime defense industry, and the growing coastal leisure industry. As one of the most advanced island countries in the world, UK’s maritime economy has played a significant role in its economic development in the past hundred forty years [3]. Despite the paramount social-economic value of marine waters, there is scant research and literature on characterizing and measuring a country’s or regional maritime economy [5,6].”

Before of these changes, we added the following references:

5.       Fernández-Macho, J.; Murillas, A.; Ansuategi, A.; Escapa, M.; Gallastegui, C.; González, P.; Prellezo, R.; Virto, J. Measuring the maritime economy: Spain in the European Atlantic Arc. Mar. Policy 2015, 60, 49-61.

6.        Fernández-Macho, J.; González, P.; Virto, J. An index to assess maritime importance in the European Atlantic economy. Mar. Policy 2016, 64, 72-81.

3rd Comment: The authors need to explain why they have chosen 2002 as the starting point for the analysis, since they have data on 1980 (section 3.3.) after the Chinese economy open. Equally important, the changes in importance of the regions’ economic contributions are left unexplored.

Our response: Thanks very much for your comment and suggestion. It is because the State Oceanic Administration of China has published annual reports after 2002. The data for the year 1980 was obtained from a publication by Xu (our Reference 10 in the revised manuscript) entitled “Developing Maritime-related Engineering Technologies to Exploit Maritime Resources” in 2002. To clarify this situation, we wrote the following sentence in Section 2.2. Data Collection:

“…It should be noted that the State Oceanic Administration of China has published annual reports since 2002…”

Regarding the regions’ economic contributions to China’s total maritime economy, there are very limited data available from the annual reports of the State Oceanic Administration of China. Thus, we decided not to over interpret the changes of region’s contributions in the manuscript. We sincerely hope that you understand the situation.

4th Comment: While the method used seems to fit well with the evolution of Chinese maritime economy, the authors would benefit by discussing this approach in relation to others used in the literature or to previous work on the Chinese case (for instance, input-output analysis, previous research on Chinese maritime sector and to other cases, for instance Korea or Japan as relevant neighbouring countries).

Our response: Thanks so much for your comment and suggestion. After reading your comments, we conducted another round of literature review and we expanded the 2nd paragraph of Introduction as:

“…Sun et al. [15] explored the growth of China’s maritime economy during the period of 1996-2013. They reported that China’s total maritime economy had growth continuously and Shanghai and Tianjin had experienced much faster development of the maritime economy than other provinces. More recently, Wang and Wang [16] studied the contribution of China’s maritime sectors to China’s total maritime economy using the input-output approach for the years 2002, 2007, and 2012, respectively. The input-output approach is able to reveal the inter-industry linkage effects by using the Leontief inverse matrix of an input-output table [16]. This approach has been used to study the forward linkage effect of different maritime sectors on the national economy in Korea [17,18] and Japan [19] for a particular year. However, there is still a lack of research studies on modeling China’s maritime economic growth based on long-term time series and exploring the growth patterns and trends of the primary, secondary, and tertiary sectors of China’s maritime economy. Wilkinson et al. [20] suggest that “long-term” means more than ten years, “medium-term” means three to ten years, and “short-term” means less than three years. Thus, the main purpose of the study is to address such gaps by applying logistic models to characterize China’s total and sectoral maritime economic growth during the period of 2002 to 2017, and to project short-term growth in China’s total maritime economy…”

Because of these changes, we added the following references:

15.      Sun, C.; Li, X.; Zou, W.; Wang, S; Wang, Z. Chinese marine economy development: Dynamic evolution and spatial difference. Chinese Geogr. Sci. 2018, 28(1), 111-126.

16.      Wang, Y.; Wang, N. The role of the marine industry in China’s national economy: An input–output analysis. Mar. Policy 2019, 99, 42-49.

17.      Kwak, S.J.; Yoo, S.H.; Chang, J.I. The role of the maritime industry in the Korean national economy: An input–output analysis. Mar. Policy 2005, 29(4), 371-383.

18.      Lee, M.K.; Yoo, S.H. The role of the capture fisheries and aquaculture sectors in the Korean national economy: An input–output analysis. Mar. Policy 2014, 44, 448-456.

19.      Nakahara, H. Economic contribution of the marine sector to the Japanese Economy. Trop. Coast 2009, 16(1), 49-53.

5th Comment: The sustainability implications are drawn from secondary sources. It would be good to know whether the discussion is based on hypothetical implications or actual findings for China’s maritime sector.

Our response: Thanks very much for your comment. You are right. Many sustainability implications are drawn from secondary sources including those publications from the World Bank and the Food and Agriculture Organization of the United Nations. Yet, the model we established is robust and the predictions should be accurate. Hence, the growth in near term will cause more consumption of natural resources and bring more impact such as greenhouse gases emissions on the environment. In fact, we added the following sentence to this section as:

“…Specifically, China’s international sea freight – a subsector of China’s maritime economy and its greenhouse gases emissions have followed logistic growth in the past decades [29]…”

It is a publication from our research showing the logistic growth of China’s international sea freight – a subsector of China’s maritime economy causing more GHG emissions in the near future.

6th Comment: In relation to the conclusions, if the authors want to relate their findings to management, more should be known about the actual policies to promote the changes. Relatedly, that would bring more insights onto causes of growth and hence more robust evidence on the strength of their predictions. Would the authors indicate that the sustainability of the sector is exclusively dependant on natural/ecological constraints? 

Our response: Thanks very much for your comment and suggestion. Due to insufficient data about resource consumption (although we had time-series data about China’s total and sectoral maritime economy from the publications of the State Oceanic Administration of China), we decided not to make any strong conclusion about what the government should do in terms of policy formulation. Nevertheless, we did provide some insights and suggestions in Sustainability Implications which is part of Discussion. We sincerely hope that you understand the situation.

7th Comment: The paper is well written and structured. It describes the changes in composition of the sector and attempts to predict. It might benefit from setting some expectations prior to the analysis as to make the discussion of the implications more insightful.

Our response: Thanks for your comment and suggestion. We made quite a bit change in Introduction that will position the objective or goal of the study more clearly. Hence, the section of Discussion (and its subsection – Sustainability Implications) should be more relevant.

Once again, please accept our sincere thanks for your valuable comments and suggestions that improve the manuscript substantially.

Reviewer 4 Report

The use of a logistic model is a good way to predict the nearest future of maritime areas in China. As consequence, looking at facts and sheets, it could be possible to discover a right approach, in order to set up a sustainability-oriented policy.

Therefore the paper pt on the table an interesting case of study, but it is important to enforce the scientific dimension of the approach.

The description of seminal aspects of maritime economy sector is clear and is strengthened by methods and data. It is not the same as regards environmental issues. Therefore

1)

could be useful to add consideration about:

- decrease of fish population, or wetlands, river, forestry due to tourism activity).

- pollutants

- bio potential

- soil-take in the coastal areas

and so on.

2)

There are many works regarding maritime activity and sustainable development, all over the world that could be quoted.

3)

Data could be also useful to understand if policies reminded by authors (such as the National Biodiversity Conservation Action Plan or the Marine Environment Protection Law Fisheries Law) gained their scopes to conserve marine environment, resources and habitats.

4)

Some reflections about the importance of the impacts, more than the aims of policies could be important.

5)

Finally, some references about these last aspects could be helpful

Author Response

Reviewer 4

1st Comment: The use of a logistic model is a good way to predict the nearest future of maritime areas in China. As consequence, looking at facts and sheets, it could be possible to discover a right approach, in order to set up a sustainability-oriented policy. Therefore the paper put on the table an interesting case of study, but it is important to enforce the scientific dimension of the approach.

Our response: Thanks very much for your comment and support.

2nd Comment: The description of seminal aspects of maritime economy sector is clear and is strengthened by methods and data. It is not the same as regards environmental issues. Therefore, it could be useful to add consideration about:

- decrease of fish population, or wetlands, river, forestry due to tourism activity).

- pollutants

- bio potential

- soil-take in the coastal areas

and so on.

Our response: Thanks so much for your comment and suggestion. In the revised manuscript, we rewrote the fifth sentence as:

“… On the other hand, coastal tourism adversely affects the biophysical environment. The impacts include the increase in air and noise pollution from increased traffic, the increase in water pollution and waste and the adverse change in wetlands and rivers due to a wide range of tourism-related commercial activities, and the decrease of fish population due to recreational fishing [34-37].”

3rd Comment:  There are many works regarding maritime activity and sustainable development, all over the world that could be quoted.

Our response: Thanks very much for your comment and suggestion. We conducted another round of literature review and included the following articles as references in the revised manuscript.

36.          Xiao, Z.; Lam, J.S.L. A systems framework for the sustainable development of a Port City: A case study of Singapore's policies. Res. Transp. Bus. Manage. 2017, 22, 255-262.

37.          Tan, W.J.; Yang, C.F.; Château, P.A.; Lee, M.T.; Chang, Y.C. Integrated coastal-zone management for sustainable tourism using a decision support system based on system dynamics: A case study of Cijin, Kaohsiung, Taiwan. Ocean Coast. Manage. 2018, 153, 131-139.

46.          Yang, L.; Wang, P.; Cao, L.; Liu, Y.; Chen, L. Studies on charges for sea area utilization management and its effect on the sustainable development of marine economy in Guangdong province, China. Sustainability 2016, 8(2), 116.

4th Comment: Data could be also useful to understand if policies reminded by authors (such as the National Biodiversity Conservation Action Plan or the Marine Environment Protection Law Fisheries Law) gained their scopes to conserve marine environment, resources and habitats.

Our response: Thanks very much for your comment and suggestion. We added the following sentences to Section 4.1:

“…Specifically, Mu et al. [45] showed that China’s marine catch had increased sharply from about 3.8 Mt in 1985 to 11 Mt in 1995. However, China’s National Biodiversity Conservation Action Plan slowed down the growth of marine catch significantly, causing the total marine catch increasing slowly to 14.2 Mt in 2002 [41,45].”

Because of the change, the following reference was added:

45.          Mu, Y.; Yu, H.; Chen, J.; Zhu, Y. A qualitative appraisal of China’s efforts in fishing capacity management. J. Ocean U. China 2007, 6(1), 1-11.

5th Comment: Some reflections about the importance of the impacts, more than the aims of policies could be important.

Our response: Thanks very much for your comment and suggestion. In the revised manuscript, we rewrote some sentences as:

“…Besides, the overexploitation and the degradation of fishery habitats have caused grave concern for both China’s government and fishermen [45]. Thus, China imposes an annual fishing moratorium between 1 May and 16 August in South China Sea, East China Sea, Yellow Sea, and Bohai Sea [45]….”

6th Comment: Finally, some references about these last aspects could be helpful.

Our response: Thanks for your suggestion. We added a number of references in the revised manuscript. These new references include:        

36.          Xiao, Z.; Lam, J.S.L. A systems framework for the sustainable development of a Port City: A case study of Singapore's policies. Res. Transp. Bus. Manage. 2017, 22, 255-262.

37.          Tan, W.J.; Yang, C.F.; Château, P.A.; Lee, M.T.; Chang, Y.C. Integrated coastal-zone management for sustainable tourism using a decision support system based on system dynamics: A case study of Cijin, Kaohsiung, Taiwan. Ocean Coast. Manage. 2018, 153, 131-139.

44.          Zou, K. Implementing marine environmental protection law in China: Progress, problems and prospects. Marine Policy 1999, 23(3), 207-225.

45.          Mu, Y.; Yu, H.; Chen, J.; Zhu, Y. A qualitative appraisal of China’s efforts in fishing capacity management. J. Ocean U. China 2007, 6(1), 1-11.

46.          Shen, G.; Heino, M. An overview of marine fisheries management in China. Mar. Policy 2014, 44, 265-272.

47.          Yang, L.; Wang, P.; Cao, L.; Liu, Y.; Chen, L. Studies on charges for sea area utilization management and its effect on the sustainable development of marine economy in Guangdong province, China. Sustainability 2016, 8(2), 116

Dear Reviewer 4 - please accept our sincere thanks for your comments and suggestions that help us improve the manuscript continually.

Round 2

Reviewer 1 Report

The authors accepted almost all of the recommendations of all reviewers, so in my opinion the authors made a important improvements based on the first reviews. The authors fullfil the areas which were missing and reconnected the topic with sustainability and economical point of view.

After analyzing the review of the paper as well as all the changes introduced during the revision process I am convinced that the review has dramatically improved the quality of the manuscript since the original submission. In fact, the paper has addressed the comments of the reviewer. On the basis of these observations, this manuscript is recommended for publication.

Author Response

1st Comment:

The authors accepted almost all of the recommendations of all reviewers, so in my opinion the authors made important improvements based on the first reviews. The authors fulfill the areas which were missing and reconnected the topic with sustainability and economical point of view.

Our response: Thanks very much for your comment.

2nd Comment: After analyzing the review of the paper as well as all the changes introduced during the revision process I am convinced that the review has dramatically improved the quality of the manuscript since the original submission. In fact, the paper has addressed the comments of the reviewer. On the basis of these observations, this manuscript is recommended for publication.

Our response: Thanks for your comment and recommendation.

Reviewer 2 Report

I appreciate all revisions you made in the first round, you tried to improve it through literature review and theory description. However, we could not detect any model improvement which is one of the key elements. It is right that there is a close connection between maritime sector and sustainability, what we are expecting in this research is a direct empirical research in model construction, 

Unfortunately, we could not detect any model improvement which was emphasized in last revision.

I understand that you intended to connect the idea of sustainability to your research, however, neither your model and data could support this intention, therefore, I would suggest you try another journal which is more suitable to your theme, or improve your model and data.

Besides, I went through other reviewers’ opinions, for example, the first comments of reviewer 4, “it is important to enforce the scientific dimension of the approach”, to my understanding, it is supposed to encourage you to improve your model, but your answer was not quite applicable.

Author Response

1st Comment: I appreciate all revisions you made in the first round, you tried to improve it through literature review and theory description. However, we could not detect any model improvement which is one of the key elements. It is right that there is a close connection between maritime sector and sustainability, what we are expecting in this research is a direct empirical research in model construction. Unfortunately, we could not detect any model improvement which was emphasized in last revision.

Our response: Thanks for understanding our effort. We would try our best to include sustainability data and a model that links China’s maritime economy to sustainability issues in this round of revision.

2nd Comment: I understand that you intended to connect the idea of sustainability to your research, however, neither your model and data could support this intention, therefore, I would suggest you try another journal which is more suitable to your theme, or improve your model and data. Besides, I went through other reviewers’ opinions, for example, the first comments of reviewer 4, “it is important to enforce the scientific dimension of the approach”, to my understanding, it is supposed to encourage you to improve your model, but your answer was not quite applicable.

Our response: Thanks for your comment. After exploring more from the data set provided by the Food and Agriculture Organization of the United Nations, we eventually obtained a strong relationship between China’s maritime economy and the exploitation of ecological resources, in this case, total fisheries production. As suggested by you, we presented a new section of 3.4 showing this specific relationship (then we elaborated the relationship in this section and Discussion) as.

               “3.4. A Model Linking China’s Maritime Economy to Its Sustainability Implication

The continual development of China’s maritime economy inevitably requires the exploitation of more ecological resources [31] and produces more pollutants such as greenhouse gases [29]. Specifically, China’s total fish production is one of the major maritime industries in China because it employs more than 14 million people in China [31]. Yet, China’s total fish production had increased by more than 82.6 percent from 38.3 million tonnes in 2002 to 70.0 million tonnes in 2017 [31]. Figure 9 shows that China’s maritime economy was plotted against China’s total fish production during the period of 2002-2017. Mathematically speaking, the linear model between China’s maritime economy and total fish production is given as follows:

                                                            (with a new equation – Equation 9)

where ytotal fish production is China’s total fish production in million tonnes and xmaritime economy is China’s total maritime economy in billion RMB. The R2 value between China’s total fish production and China’s total maritime economy was 0.9977, meaning that more than 99.77 percentage change in variance of China’s total fish production could be explained by change in China’s total maritime economy. Equation (9) implies that a change of 1,000 billion RMB will lead to an increase of 4.7 million tonnes of total fish production. As China’s total maritime economy would increase by 1,133 billion RMB in from 2017 to 2019 (see Section 3.1), it is expected that this change will lead to China’s total fish production increasing by another 5.3 million tonnes. This potential change in total fish production on one hand will cause a greater ecological impact, and on the other hand, can further strain China’s relationship with its neighboring countries such as Japan, Korea, Vietnam, and Thailand due to fishing in open seas [32].

                                                            (with a new figure – Figure 9)

Figure 9. Change in China’s total fish production vs. total maritime economy from 2002 to 2017.”

Because of this new subsection, we added a new reference:

Mallory, T. G. China’s distant water fishing industry: Evolving policies and implications. Mar. Policy 2013, 38, 99-108.

Once again, please accept our sincere thanks for pointing this research direction to us.

Reviewer 4 Report

Many thanks for your answer. I think that now the paper could be published

Author Response

1st Comment: Many thanks for your answer. I think that now the paper could be published.

Our response: Thanks so much for your comment and recommendation.

Round 3

Reviewer 2 Report

thank you for your work and congratulations